# Philadelphia-Negative Chronic Myeloproliferative Neoplasms during the COVID-19 Pandemic: Challenges and Future Scenarios

**DOI:** 10.3390/cancers13194750

**Published:** 2021-09-23

**Authors:** Francesca Palandri, Massimo Breccia, Valerio De Stefano, Francesco Passamonti

**Affiliations:** 1IRCCS Azienda Ospedaliero-Universitaria di Bologna, Istituto di Ematologia “Seràgnoli”, 40138 Bologna, Italy; francesca.palandri@unibo.it; 2Department of Translational and Precision Medicine, Azienda Ospedaliera Policlinico Umberto I, Sapienza University, 00185 Rome, Italy; breccia@bce.uniroma1.it; 3Section of Hematology, Department of Radiological and Hematological Sciences, Catholic University School of Medicine, Fondazione Policlinico Universitario Agostino Gemelli IRCCS, 00168 Rome, Italy; 4Department of Medicine and Surgery, University of Insubria, ASST Sette Laghi, 21100 Varese, Italy; francesco.passamonti@uninsubria.it

**Keywords:** coronavirus, COVID-19, MPNs, cancer, pandemic

## Abstract

**Simple Summary:**

Little information has been reported about the impact of the COVID-19 pandemic in Philadelphia-negative chronic myeloproliferative neoplasms (MPN). In this review, we summarize the knowledge about MPN clinical management, including cytoreductive and antiplatelet/anticoagulant therapy, thrombotic risk, prognosis, and vaccination strategies at the time of COVID-19.

**Abstract:**

An outbreak of severe acute respiratory syndrome coronavirus 2 (SARS-CoV2) started in December 2019 in China and then become pandemic in February 2020. Several publications investigated the possible increased rate of COVID-19 infection in hematological malignancies. Based on the published data, strategies for the management of chronic Philadelphia-negative chronic myeloproliferative neoplasms (MPNs) are provided. The risk of severe COVID-19 seems high in MPN, particularly in patients with essential thrombocythemia, but not negligible in myelofibrosis. MPN patients are at high risk of both thrombotic and hemorrhagic complications and this must be accounted in the case of COVID-19 deciding on a case-by-case basis. There are currently no data to suggest that hydroxyurea or interferon may influence the risk or severity of COVID-19 infection. Conversely, while the immunosuppressive activity of ruxolitinib might pose increased risk of infection, its abrupt discontinuation during COVID-19 syndrome is associated with worse outcome. All MPN patients should receive vaccine against COVID-19; reassuring data are available on efficacy of mRNA vaccines in MPNs.

## 1. Introduction

### 1.1. Overview of Coronavirus Pandemic

Alpha and beta Coronaviruses (CoVs) are a subfamily of large and enveloped viruses that are known to infect humans, mainly through respiratory transmission [1,2]. Severe acute respiratory syndrome coronavirus (SARS-CoV) and Middle East respiratory syndrome coronavirus (MERS-CoV) have caused widespread concern resulting in epidemics with significant morbidity and mortality in 2002 and in 2012, respectively [3,4].

In December 2019, a novel coronavirus was identified to be responsible for unidentified pneumonia outbreaks in Wuhan, Hubei Province, China [5]. Since the genomic sequence of the current virus is closer to that of SARS-CoV than that of MERS-CoV, the nomenclature SARS-CoV-2 was given to the causative pathogen of coronavirus disease 2019 (COVID-19) [6].

The incubation period of COVID-19 is between 1 and 14 days, mostly between 3 and 7 days. Transmission occurs primarily via respiratory droplets and close contacts. The SARS-CoV-2 infection may be responsible for different outcomes ranging from asymptomatic infection (6.4%), mild to moderate cases (79.6%), severe (5.3%), critical (7.3%) and fatal cases (1.4%) [7]. Mild and moderate cases are mainly characterized by symptoms related to upper airways infection (fever, fatigue and dry cough, sore throat) and inflammation (myalgia, arthralgia and headache). Severe and critical cases are characterized by multiple complications including respiratory distress, thromboses, sepsis, acute kidney injury, acute cardiac injury and multi-organ dysfunction.

Along with clinical stages, also a progressive coagulopathy may be observed in COVID-19 patients. Stage 1 (mild infection) is characterized by mildly systemic coagulopathy. Stage 2 includes moderate and severe cases and is characterized by pulmonary inflammation and coagulopathy with localized microthrombi. Stage 3 includes critical and fatal cases and is characterized by a severe hyperimmune syndrome and systemic coagulopathy associated with thrombocytopenia and high risk of severe thrombosis (pulmonary embolism, deep vein thrombosis) [8].

### 1.2. Impact of Coronavirus Pandemic on Cancer Patients

With around 161,310,781 infected people and over 3,347,409 deaths registered worldwide between February 2020 and June 2021, the COVID-19 pandemic is an emergency of major international concern [9,10]. 

The detrimental impact of a cancer diagnosis on SARS-CoV-2 infection was rapidly observed in a study on the Chinese population, in which 1% of patients with COVID-19 had cancer, whereas the incidence of cancer was 0.29%. While epidemiological findings could be related to a closer medical follow-up in cancer patients, more severe respiratory complications were ascertained in this cohort as compared with patients without cancer (39% vs. 8%, respectively; *p* = 0.0003) and a history of chemotherapy in the month preceding infection was negatively associated with survival (odds ratio 5.34, 95% CI 1.80–16.18; *p* = 0.0026) [11,12,13]. A recent meta-analysis including 41 studies with 16,495 COVID-19 patients observed that proportion of hypertension (OR: 1.98, 95% CI: 1.62–2.42), diabetes (OR: 2.04, 95% CI: 1.67–2.50), cardiovascular disease (OR: 2.78, 95% CI: 2.00–3.86) and cancer (OR: 1.75, 95% CI: 1.40–2.18) were significantly higher in patients with severe viral infection, confirming that comorbidities and cancer may affect the outcome of SARS-CoV-2 infection [14].

Philadelphia-negative (Ph-) chronic myeloproliferative neoplasms (MPNs) are clonal disorders including polycythemia vera (PV), essential thrombocythemia (ET) and myelofibrosis (MF), primary MF (PMF) or secondary to ET or PV (PPV/PET-MF) [15]. These chronic cancers are characterized by increased thrombotic risk, progressive splenomegaly, debilitating systemic symptoms and reduced survival [16].

Infections are frequently reported in MPNs, representing the ultimate cause of death in approximately 10% of patients [17,18]. Infections are primarily bacterial (78%) but viral (11%) and fungal (2%) infections can also develop [17].

Infectious risk in MPNs is mainly caused by deregulation of key mediators of the immune system. In particular, monocytes/macrophages, T cells, natural killers and myeloid-derived suppressor cells are often characterized by numerical and/or functional abnormalities [19,20,21,22,23,24,25].

The use of agents with immunosuppressive activity, including the JAK1/2 inhibitor ruxolitinib (RUX) may further increase the risk of infections [17,26,27,28,29,30,31,32,33,34] (Figure 1).

In a short time, the global research community has made an impressive effort to report the different characteristics of COVID-19 while taking care of patients. However, information on SARS-CoV-2 infection impact on management and outcome of MPN patients is scarce. To understand these concerns, a detailed search of the literature was conducted using PubMed (US National Library of Medicine and the National Institutes of Health) and Web of Science (Thomas Reuters Online Academic Citation Index), with publication dates ranging from 2000 to June 2021. To ensure an extensive range of publications were identified, broad search terms for ET, PV, MF, COVID-19, SARS-CoV-2, vaccine and clinical/epidemiological variables (e.g., incidence, prevalence, frequency, diagnosis, pathogenesis, infections, thrombosis, bleedings, complications, survival, outcome) were utilized. Furthermore, we reviewed the literature cited in the identified papers.

Based on this research, we have outlined currently available data to answer the most frequently asked questions related to the impact of the COVID-19 pandemic on MPN clinical management, including cytoreductive and antiplatelet/anticoagulant therapy, thrombotic risk, prognosis and vaccination strategies.

## 2. Is the Risk of Infection and Severe SARS-CoV-2 Illness Higher in MPN Patients?

Infections are one of the main causes of morbidity and mortality in patients with MPNs. In a Swedish population-based study, the risk of dying of infection was 2.5-fold, 4.6-fold and 10.4-fold higher in patients with ET, PV and MF, respectively, compared to age- and sex-matched healthy controls [18]. In addition, in a German–Italian patient-reported pilot study, MF diagnosis and ruxolitinib therapy were associated with higher infectious risk [35]. Among MF patients, worse disease status in terms of higher International Prognostic Score System (IPSS) risk and large (≥10 cm below costal margin) splenomegaly were significantly associated with infectious risk [17].

### 2.1. Risk of Infection SARS-CoV-2 in MPN Patients

In 271 German MPN patients, no COVID-19 infection or positive SARS-CoV-2 test result was reported during the survey period and only 1% of people in close contact with patients tested positive for SARS-CoV-2. However, a medical mask was used by almost all patients and 26% had spontaneously quarantined themselves. The observed low incidence of COVID-19 infections in this relatively small patient population may be related to random variations in infection rates. Nonetheless, this survey suggests that adherence to basic prevention rules can reduce the risk of infection [36]. In a prospective study on 257 MPN patients from Canada, only 1% contracted COVID-19 infection, much less than would be expected given provincial infection rates [37]. In another survey conducted on 964 MPN patients from United Kingdom, 96.1% of respondents reported no previous confirmed COVID-19 infection; notably, 91.5% of respondents reported COVID-19 vaccination when they had the opportunity and only 0.8% of respondents declined offer of COVID-19 vaccination [38].

Between February and April 2020, an Italian survey involved 34 blood centers and a cohort of 13,248 Ph- MPN patients. A total of 36 patients had been infected with COVID-19 (33.6% of patients tested but 0.002% of the entire cohort). Of these, 13 (36%) were asymptomatic, 13 had had mild flu-like symptoms (36%) and 10 had developed pneumonia (four patients required invasive ventilation). The mortality rate from COVID-19 was 22% (34% of symptomatic patients) [39]. In the registry of the American Society of Hematology, 92 (9.7%) out of 947 hematology patients affected by COVID-19 syndrome had a Philadelphia-negative or positive myeloproliferative neoplasm [40].

According to the aforementioned reports, the risk of getting the SARS-CoV-2 infection seems not elevated in MPNs. However, such conclusion can be weakened by the relatively small number of patients investigated. A nation-wide database of patient electronic health records of 73 million patients in the US was analyzed for COVID-19 and eight major types of hematologic malignancies (including 121,200 ET patients and 134 72,150 PV patients, PMF was not included in the analysis). Patients with hematologic malignancies had increased odds of COVID-19 infection compared with patients without hematologic malignancies for both all-time diagnosis (adjusted odds ratio, AOR, 2.27) and recent diagnosis in the past year (AOR 11.91). ET produced the strongest effect, after acute lymphoid leukemia, for both all-time diagnosis (AOR 4.29) and recent diagnosis (AOR 20.65). PV patients had a OR for COVID-infection of 1.43 and 4.89 (all-time diagnosis and recent diagnosis, respectively [41].

### 2.2. Mortality for SARS-CoV-2 Infection in MPN Patients

In the cohort collected by the Italian Hematology Alliance on COVID-19, that included 536 patients admitted to 66 Italian hospitals between 25 February and 18 May 2020, with symptomatic COVID-19, 15% had an MPN [42]. The same study highlighted that the presence of any hematological neoplasms significantly worsened survival in case of COVID-19, with a standardized mortality ratio of 3.72 in hematological patients under 70 years of age compared to the Italian healthy population. In addition, having COVID-19 significantly impaired survival in patients with hematological malignancies (standardized mortality ratio of 41.3 compared to hematological non-COVID-19 population) [42].

A population-based registry study from Spain collected 883 patients with hematologic malignancies and COVID-19 syndrome. Patients with acute myeloid leukemia (AML, 2.22 versus non-Hodgkin’s lymphoma, NHL) and active antineoplastic treatment with monoclonal antibodies (2.02) were associated with higher mortality. In contrast, but not surprisingly, lower mortality was observed in patients with Ph-negative MPN compared to AML and NHL patients (0.33) [43]. In a survey on 77 patients from the United Kingdom with MPN and COVID-19 (82% hospitalized, 35% with MF), the case fatality rate among the inpatients was 52%. In a comparison cohort of 60,430 COVID-19 hospitalized patients, the rate of mortality was 28% [44].

In the European Leukemia Net (ELN) International Study on MPN and COVID-19, 175 MPN patients (PV *n* = 46; ET *n* = 51; pre-fibrotic MF *n* = 18; PMF *n* = 60) who developed COVID-19 from 15 February to 31 May 2020 were collected in 37 European hematology Centers. During the acute phase of the infection, in-hospital mortality affected 27.4% of patients and the most vulnerable MPN subgroup was overt PMF (mortality 48%); notably, the proportion of patients admitted to Intensive Care Unit (ICU) was 10.9%. The diagnosis of ET, PV and pre fibrotic-PMF did not influence the proportion of survivors versus non-survivors [45].

Finally, a recent review of 13 cohort or population studies reported that patients with a hematological malignancy, especially those diagnosed recently and with myeloid neoplasia including myeloproliferative disorders, are at increased risk of death with COVID-19 compared to the general population [46].

Post-COVID-19 related consequences including vascular complications and clonal evolution into MF, myelodysplasia (MDS) and acute myeloid leukemia (AML) were also explored in 125 of the 175 patients (71%) enrolled in MPN-COVID ELN study, who survived to the acute phase of infection [47]. Notably, deaths occurred in eight patients after 9 months, with a 9% probability of death. The event-free survival (thrombosis, cancer and death) was 66% in the 125 patients followed for a median of 6 months post-COVID-19. Overall, these data indicate that the health consequences of COVID-19 extend beyond acute infection and suggest careful surveillance in all patients with MPNs.


*Compared to the healthy population, MPN patients are at higher risk for all infections. The likelihood of having COVID-19 seems higher in MPNs. MF are less able to recover from COVID-19. MPN patients require aggressive infection prevention strategies with strict adherence to coronavirus safety protocols. There is some evidence that this policy can significantly reduce the risk of infection. After COVID-19 infection, MPN patients should receive adequate clinical and laboratory monitoring.*


## 3. Should Antiplatelet or Anticoagulant Therapy Be Changed in MPN Patients during the SARS-CoV-2 Pandemic?

A recent international survey analyzed 442 MPN patients receiving direct oral anticoagulants (DOAC) because of a concomitant diagnosis of atrial fibrillation or a history of venous thromboembolism (VTE) [48]; the overall rate of thrombosis and major bleeding was comparable to that previously reported in MPN patients receiving vitamin K antagonists (VKA) [49], furnishing indirect evidence of a similar efficacy and safety. Major bleeding was more frequent in patients receiving dabigatran or with diagnosis of MF [49]. Therefore, DOAC could represent a possible alternative to VKA for antithrombotic prophylaxis given the advantage in ease of administration and improved patient convenience. In the pandemic scenario, they can represent an alternative to VKA to reduce the need for hospital visits for INR control.

The COVID-19 syndrome is known to be associated with coagulation disorders and increased risk of vascular events [50]. COVID-19-associated coagulopathy has prompted the use of standard supportive care measures and thromboembolic prophylaxis for critically ill hospitalized patients [51]. A multicenter retrospective study on 400 hospital-admitted COVID-19 patients receiving standard-dose prophylactic anticoagulation observed an overall thrombotic complication rate of 9.5%, while the overall and major bleeding rates were 4.8% and 2.3%, respectively [52].

In a cohort of 162 MPN patients with SARS-CoV-2 infection (PV *n* = 42; ET *n* = 48; prefibrotic MF *n* = 16; PMF *n* = 56), 15 major thromboses (12 venous, 8 of which occurred in patients with ET) were collected. All but one patient were receiving low molecular weight heparin (LMWH) prophylaxis. The cumulative incidence of arterial and venous thromboembolic events adjusted for risk of death was 8.5% after 60 days of observation. At diagnosis of COVID-19, platelet counts were significantly lower (*p* < 0.0001) than at the last pre-COVID follow-up. The decline in platelet count was significantly higher in ET (−23.3%, *p* < 0.0001) than in PV (−16.4%, *p* = 0.1730) and was associated with a higher mortality rate (*p* = 0.0010) from pneumonia. Independent risk factors for thrombosis were ICU transfer (SHR = 3.73, *p* = 0.029), neutrophil/lymphocyte ratio (SHR = 1.1, *p* = 0.001) and ET phenotype (SHR = 4.37, *p* = 0.006) [53]. Seven out of 162 patients (4.3%) developed major bleeding, particularly in the MF; these events were diagnosed later, i.e., starting 7–10 days after the onset of SARS-CoV-2 infection. Overall, the rate of thromboses seem to be higher in patients with ET that acquire SARS-CoV-2 infection (16.6%), while in MPN-COVID patients, the rate of bleeding is high, but overall comparable to newly diagnosed non-COVID MPN patients and to non-MPN COVID patients [52,54,55,56].

In line with the recommendations contained in the latest ASH version of the international panel of MPN experts [57], all hospitalized MPN-COVID patients should receive prophylactic doses of LMWH. Operationally, the treatment of pre-fibrotic myelofibrosis does not differ much from that of ET [58] and the thrombotic risk is similar [59].

High-risk condition concerns hospitalized patients with MPN treated with both VKA or DOACs for pre-COVID-19 chronic atrial fibrillation or history of venous thromboembolism (VTE). Given the possible metabolic interaction of these drugs with most antiretroviral drugs on liver cytochromes, such as CYP2C9 and CYP3A4 [60], the advice is to replace oral anticoagulants with LMWH prophylaxis in hospitalized MPN-COVID patients [57].

In patients with pre-COVID-19 arterial thrombosis (transient ischemic attack, ischemic stroke, myocardial infarction, peripheral arterial thrombosis) or with percutaneous coronary intervention (PCI) (within ≤3 months), it is strongly recommended not to discontinue antiplatelet drugs, unless clinical circumstances or hemorrhagic events prevent it [61]. 


*There is no indication that patients with MPN should change antiplatelet or anticoagulant therapy if SARS-CoV-2 negative. For MPN patients with a new event requiring oral anticoagulation, DOAC instead of VKA might be taken into consideration to reduce in-hospital visits.*



*MPN patients are at high risk of both thrombotic and hemorrhagic complications and this must be accounted in the case of COVID-19 deciding on a case-by-case basis and considering overall performance status, thrombotic/hemorrhagic risk and laboratory/hematology parameters. Switch from oral anticoagulation to LMWH may be considered in SARS-CoV-2 positive MPNs as platelet count can reduce promptly and for a lesser drug-drug interaction. The use of LMWH is recommended in all hospitalized COVID-19 MPN patients, in replacement of ongoing low-dose aspirin. LMWH should be added to ongoing aspirin in the case of pre-COVID-19 history of arterial thrombosis.*



*ET patients are at higher risk of thrombosis and need special clinical surveillance; the benefit of a combined treatment of LMWH and aspirin, given the possible role of platelets, should be investigated by ad hoc studies.*


## 4. Should Cytoreductive Therapy or Phlebotomies Management Be Changed during the SARS-CoV-2 Pandemic?

In ET and PV, the major goal of therapy is to reduce thrombotic and hemorrhagic events and to monitor for disease progression including transformation into acute leukemia and myelofibrosis [62].

All patients with PV require phlebotomy to keep hematocrit (Hct) below 45% and once-daily low-dose aspirin, in the absence of contraindications [63]. There is no evidence that phlebotomy indications should be modified because of the COVID-19 pandemic. Strategies that may reduce the need for phlebotomies (i.e., increased water intake and/or start of cytoreduction) and in-hospital visits (i.e., decreased waiting time for phlebotomy through an online ticketing system; telehealth, rapid vaccination of health care professionals; increasing disinfection of all the contacting surfaces) should be implemented [64,65,66].

Cytoreductive therapies are indicated in PV and ET patients at high thrombotic risk of (age ≥60 years and/or history of thrombotic event) and in selected low-risk patients [67,68,69].

Hydroxyurea/hydroxycarbamide (HU) is the most widely used cytoreductive agent in PV and ET [70,71,72]. Among HU-intolerant patients, less than 5% develop a hematological toxicity (namely, absolute neutrophil count <1.0 × 10^9^/L or platelet count <100 × 10^9^/L or hemoglobin <10 g/dL at the lowest dose of HU required to achieve a complete or partial clinico-hematologic response) [73]. In the Continuation-PV trial that randomized early-stage PV patients to receive either hydroxyurea or ropeg-interferonα2b, 2% of both HU and interferon-treated patients had a grade 3 neutropenia without grade 4 events, which was reversible after drug temporary discontinuation or dose reduction [74].

Overall, immunosuppression related to HU therapy is infrequent, transitory and with no clinical relevance. Additionally, HU has an immunomodulatory effect in sickle cell anemia (SCA), which is, like COVID-19, a hyperinflammatory thrombogenic syndrome. The cytostatic effect of HU on CD4 and CD8 T cells may decrease the abnormal production of proinflammatory cytokines during COVID-19, reducing the severity of clinical symptoms. HU was also shown to exert an antiviral effect in human immunodeficiency virus (HIV) infections [75]. In vitro, HU inhibits viral DNA synthesis studies of HIV-infected lymphocytes and has a synergic activity with nucleoside reverse transcriptase inhibitors. Possibly, HU may also in COVID-19 attenuate viral load by decreasing CD4 T cell proliferation and preventing the exhaustion of CD8 T cells [76,77,78].

A recent Italian survey showed that HU was started in all ET and PV patients at high thrombotic risk by 82.6% of hematologists. Nonetheless, HU was started only if also cardiovascular risk factors were present in high-risk patients by 13% of treating hematologists [79].

Interferons (IFNs) has been shown to have therapeutic activity against MPNs and are indicated in the first and second-line therapy of ET and PV [16,80,81].

To date, no evidence of association between IFN therapy and the clinical course of SARS-CoV-2 infection in MPN patients has been reported. Rather, type 1 IFNs induce cell-autonomous antiviral immunity and their levels dramatically increase in response to viral infections [82]. Due to their broad antiviral activity, they are currently being tested for the treatment of early-stage COVID-19 infections [83,84].

Nonetheless, more than 50% of the Italian hematologists who responded to a GIMEMA survey declared to postpone IFN start after the resolution of the pandemic, possibly due to an increased number of blood tests and hematological visits required during the first period of treatment [79]. Conversely, during the pandemic most hematologists did not change the treatments that were already ongoing. Only a minority of the hematologists discontinued HU or IFN (2.2% and 5.6%, respectively) and only 4.3% and 5.6% decreased their doses [79].

PV patients treated with phlebotomies should maintain their hematocrit target <45%: any effort should be done to maintain this approach during the pandemic.


*There are currently no data to suggest that hydroxyurea increases the risk of COVID-19. Therefore, it is believed that therapy should not be modified with dose reductions. On the contrary, it is considered appropriate to continue therapy to reduce the risk of frequent thrombotic events with COVID-19. In this sense, the use of hydroxyurea as prevention of thromboses has also been considered in other diseases such as sickle cell anemia during COVID-19 infection [85].*



*In patients that are under interferon therapy, no therapeutic modification is necessary during the COVID-19 pandemic since interferon does not increase the risk of getting SARS-CoV-2 infection. Even in case of overt COVID-19 infection, there is no evidence that this therapy should be modified.*


## 5. Ruxolitinib Use: From MPN Therapy to Control of SARS-CoV-2 Hyperimmune Syndrome

Ruxolitinib (RUX) is the first-in-class JAK1/2 inhibitor and represents the standard front-line therapy for MF-related splenomegaly and symptoms. RUX is also approved for the treatment of inadequately controlled PV after hydroxyurea failure because of intolerance or resistance [86,87,88,89,90,91,92,93].

JAK1/2 inhibition decreases pro-inflammatory cytokines which causes the improvement of disease-related symptoms but also impairs immune function. Indeed, RUX use is associated with a number of abnormalities of adaptive and innate immunity [94] (Figure 1). Clinically, opportunistic and atypical infections have been described during RUX therapy [17,26,95,96,97,98,99,100,101,102,103,104,105]. Therefore, a systematic infectious screening is recommended before the start of ruxolitinib [106].

Ruxolitinib-induced decreased immunosurveillance was initially considered a risk factor for an increased chance of acquiring SARS-CoV-2 infection and/or developing a more severe COVID-19 syndrome in patients with MPNs [57]. However, subsequent evidence has demonstrated that RUX administration was not associated with reduced survival in patients affected by COVID-19 infections. Conversely, patients who discontinued RUX had a significantly worse prognosis compared to COVID-19 MPN patients that could continue RUX therapy during the infection. Ruxolitinib discontinuation is associated with an 8.51-fold increased risk of death at multivariable analysis (*p* = 0.037) [39].

Such data strongly suggests that discontinuation of ruxolitinib in the setting of COVID-19 infection may be deleterious and should be avoided if clinically feasible [57].

In a recent GIMEMA survey, most hematologists declared to have started RUX according to routine practice during the first pandemic wave, particularly in patients with MF. Before ruxolitinib start, 40.2% of the hematologists obtained a negative COVID-19 pharyngeal swab, while in case of flu-like symptoms, COVID-19 swab is performed by most hematologists. Overall, most (79.8%) of the hematologists believed that ruxolitinib had no negative effect on COVID-19 infection, while 10.1% believed that the drug could have a negative influence in patients with MF and/or a more severe disease status. Conversely, 10.1% of hematologists anticipated a negative effect in all patients. In case of mild and moderate COVID-19 infection, 67% and 58.4% of treating hematologists would not stop or reduce ruxolitinib, respectively [79].

On the other hand, the ability of RUX to reduce the production of pro-inflammatory cytokines might have a beneficial effect on the course of COVID-19. Indeed, SARS-CoV-2 infection is associated to a cytokine storm syndrome triggered by dysregulated immune responses [107]. The cytokine storm includes a high inflammatory response with elevated levels of cytokines and immune cells that may cause organ dysfunction and in particular lung lesions, respiratory distress, multiple organ damage and death [108]. Cytokines regulate several cellular and immune processes controlled by the JAK/STAT pathway [109]. The IL6/JAK/STAT3 signaling pathway is a specific branch of the JAK/STAT pathway; IL6 is an essential pleiotropic cytokine produced by B cells, T cells, dendritic cells and macrophages able to generate an immune response or inflammation [110]. In the COVID-19 cytokine storm, IL6 is one of the most highly expressed cytokines: one of the main indicators of poor prognosis in SARS-CoV-2 infection is represented by elevated serum levels of IL6 [111].

As a result, different therapeutic strategies to treat COVID-19 related hyperinflammation include the use of JAK/STAT inhibitors (Table 1) [112]. However, the randomized phase III RUXCOVID study evaluating ruxolitinib on top of standard of care therapy in COVID-19 patients was prematurely closed, since it did not meet its primary endpoint (reduction of the number of hospitalizations for COVID-19).


*In MPN patients who are SARS-CoV-2 negative during the pandemic, ruxolitinib should be started or continued with no modifications according to guidelines. A negative SARS-CoV-2 swab test is generally not requested before ruxolitinib start; however, obtaining the COVID-19 status might be useful. There is also no indication to modify ruxolitinib doses in patients with asymptomatic/mild COVID-19. However, a dose reduction of ruxolitinib may be temporarily performed to reduce drug–drug interactions worsening hematology parameters or patients’ clinical status. To note, the discontinuation of ruxolitinib in MPNs with COVID-19, outside a clear indication, seems associated to a worse outcome and is discouraged [39].*


## 6. Prevention of SARS-CoV2 Infection in MPN Patients: What Vaccines to Use, When and with What Precautions?

Vaccines are considered the most promising approach to prevent SARS-CoV-2 infection and control the pandemic [113]. SARS-CoV-2 genome contains single-stranded positive-sense RNA encapsulated within a membrane envelop with an average diameter of 75–150 nm [114]. The SARS-CoV-2 spike (S) surface glycoprotein is a large highly antigenic type I transmembrane protein with the ability to induce the humoral and cellular immune responses [115,116]. Vaccines for SARS-CoV-2 include live attenuated vaccines, inactivated vaccines, recombinant protein vaccines, vector vaccines, deoxyribonucleic acid (DNA) vaccines and messenger ribonucleic acid (mRNA) vaccines (Table 2) [117,118,119,120,121,122,123,124,125,126,127].

According to the Danish National Patient Registry, the reported incidence rate of thromboembolic events among vaccinated Europeans is not increased relative to the expected number estimated from incidence rates from the entire Danish population [128]. However, this Danish report could not rule out a causative relationship between vaccines and thrombotic events [129]. Thirty-nine cases of vaccine-induced immune thrombotic thrombocytopenia (VITT), a syndrome characterized by thrombosis and thrombocytopenia that developed soon after vaccination with the chimpanzee adenovirus ChAdOx1 nCoV-19 vector (AstraZeneca), have been described [130,131,132]. Patients were more frequently young (<50 years) females and events mainly occurred at atypical sites (cerebral venous sinus or portal/splanchnic/hepatic veins thromboses) accompanied by low platelet count and high levels of antibodies to platelet factor 4 (PF4)–polyanion complexes despite the absence of heparin [132]. Successively 12 cases of cerebral venous sinus thrombosis meeting the clinical features of VITT have been reported after vaccination with the human adenovirus Ad26 vector (Janssen/Johnson & Johnson) [133]. Patients’ ages ranged from 18 to younger than 60 years; all were white women. So far only one case of possible VITT has been reported after the second dose of a mRNA vaccine (Moderna) in a 65 y.o. man with multiple vein thromboses, including cerebral veins and thrombocytopenia [134]. However, the number of Moderna COVID-19 vaccine doses administered in the United States as of 22 July 2021 was 137 million (source: Centers for Disease Control and Prevention) and the risk of VITT should be considered associated only to viral vector-based vaccines. More than one hundred cases of VITT syndrome have been described in detail [135]. The incidence of VITT syndrome estimated by the reports of the UK, Europe and US Regulatory Agencies is 1.3 per 100,000 first doses and 1.3 per million second doses of the Astrazeneca vaccine; the incidence of AD26.CoV2.S vaccine (Janssen/Johnson & Johnson, manufactured by Janssen Biotech, Inc., a Janssen Pharmaceutical Company of Johnson & Johnson, Horsham, PA 19044, USA) is 3.2 per million doses [136]. Due to the very low prevalence of VITT, the overall benefits of the vaccine in preventing COVID-19 outweigh the risks of side effects [137,138].

COVID-19 elicits an impaired antibody response against SARS-CoV-2 in patients with hematological malignancies [139]. Hence, patients with hematological neoplasia are likely to have lower responses to vaccines due to reduced immunological competence that is related to both the hematological disease and the immunosuppressive and/or myelotoxic effects of the treatments [140]. Particularly, patients with MPNs suffer from distinct immune deficiencies and receive different treatments that variously affect the vaccine response [141,142]. In a cohort of 30 MPN patients at 5 weeks from the administration of the BNT162b2 (Pfizer-BioNTech, manufactured by Pfizer Inc., New York, NY 10017, USA, for BioNTech Manufacturing GmbH, 55131 Mainz, Germany) vaccine, seroconversion at cutoff of 15 AU/mL Ig was reported in 88% [143]. Very recently, a memory T cell response was observed in 16 (80%) MPN patients having received a first dose of the BNT162b2 (Pfizer-BioNTech) vaccine. After 21 days, a CD4+ T cell response was observed in 15 (75%) individuals and a CD8+ T cell response was observed in seven (35%). A polyfunctional T cell response was also observed in 13 (65%) of the patients. The administration of specific therapy was not associated with significant differences in T cell or antibody responses compared to active surveillance. In addition, no significant differences were observed between patients taking ruxolitinib and those receiving other therapies [144]. On the opposite, seroconversion post-COVID-19 vaccines has been reported to be negatively affected in patients receiving ruxolitinib [145,146,147].
cancers-13-04750-t002_Table 2Table 2Main features of COVID-19 vaccines.DeveloperPlatformMechanismAdvantagesLimitationsDosesEfficacyNo. of SubjectsLocal Adverse Events (Pain, Erythema, Swelling in the Injection Side)Systemic Adverse Events (Fever, Headache, Fatigue, Myalgia, Arthralgia)Severe Adverse EventsPfizer/BioNTech [119]mRNAmRNA encoding for target viral proteinsNo interactions with the recipient’s DNATo be stored at very low temperatures2 (3 weeks apart)95.0%43,448<55y: 83%/88% (1st/2nd injection) >55y: 71%/66% (1st/2nd injection)<55y: 47%/59% (1st/2nd injection)>55y: 34%/51% (1st/2nd injection)0.6%Moderna [118]mRNAmRNA encoding for target viral proteinsNo interactions with the recipient’s DNATo be stored at very low temperatures2 (4 weeks apart)94.1%30,42084.2%/88.6% (1st/2nd injection)54.9% (1st injection) and 79.4% (2nd injection)0.5%Janssen/Johnson & Johnson [124]DNA Adenovirus vectorPlasmid DNA that contains mammalian expression promotors and the target geneHighly stableLow immunogenicity167.0%805<55y: 64%/78% (low/high-dose)>55y: 41%/42% (low/high-dose)<55y: 65%/84% (low/high-dose)>55y: 46%/55% (low/high-dose)1%/7% (low/high-dose)AstraZeneca/University of Oxford/Serum Institute of India [126]DNA AdenovirusvectorPlasmid DNA that contains mammalian expression promotors and the target geneHighly stableLow immunogenicity2 (4/8 to 12 weeks apart)70.4%11,636n.r.n.r.175 adverse events (84 in the Vaxzeria group)Novavax [148]Recombinant proteinViral proteins that have been expressed in one of various systemsSafe; no live components ofthe virusMemory is to be tested2 (3 weeks apart)89.0%131about 85%about 79%1 severe local event8 severe systemic eventsGamaleya Institute [149]DNA Adenovirus vectorsPlasmid DNA that contains mammalian expression promotors and the target geneHighly stableLow immunogenicity2 (3 weeks apart)91.6%21,9775.4%15.2%0.3%


In a small cohort of MPN patients, those receiving treatment with peg.interferon had the highest serological response in comparison with ruxolitinib or hydroxyurea. Notably, only two-thirds of the patients not receiving cytoreduction seroconverted [150].

All patients with MPN must receive vaccination against COVID-19. No specific MPN-related risks associated with vaccination are known. There are reassuring data concerning efficacy of COVID-19 mRNA vaccine in the general MPN population; however, the use of ruxolitinib could impair the seroconversion after vaccination.

## 7. What Is the Value of Telemedicine in Patients with MPNs during the SARS-CoV-2 Pandemic?

Telemedicine is the use of telecommunications technology as a tool to deliver health care [151,152,153,154,155].

The COVID-19 pandemic further highlighted how health care facilities can spread the virus and focused attention on new models of care that reduce in-person contacts to lessen the transmission of the virus and protect medical practitioners from infection [156,157,158,159,160]. Accordingly, the indication to move outpatient clinics to telephone or video-conferencing appointments was suggested in all patients with cancer, including MPNs [57,161,162].

During the first pandemic wave, the Italian GIMEMA Working Group on MPNs launched a survey that was responded by 98 Italian Hematologists with specific focus on clinical management of MPNs during the pandemic. Over 80% medical visits were converted into telehealth in 19.8%, 38% and 50% of MF, PV and ET patients, respectively [79]. In addition, hematologists declared a certain propensity to expand the use of telemedicine after pandemic resolution [79].

A recent analysis concerning the use of telemedicine at the Institute of Hematology “L. and A. Seràgnoli”, Bologna, Italy, showed that during the first pandemic wave (9 March–4 May 2020) a total of 365 out of 489 (74.6%) visits were converted to telephone appointments. Compared to patients receiving a telephone contact, the patients who required in-person visits were more frequently affected by MF (*p* < 0.001), were under active therapy (*p* = 0.03) and enrolled into a clinical trial (*p* < 0.001) [163]. Overall, 87 (23.8%) out of 365 patients who were involved in the telemedicine project also responded to a satisfaction questionnaire that analyzed 1. adequacy of medical care; 2. psychological impact of telemedicine; 3. adequacy of IT system; 4. possible advantages and future use of telemedicine. Telemedicine resulted in an overall good level of patients’ satisfaction [163]. However, all patients complained due to the lack of physical interaction, which may result in reduced diagnostic accuracy, worse symptom control, impaired evaluation of clinical signs (i.e., splenomegaly), delayed recognition and worse management of MPN-related complications. In addition, the protection of medical-legal aspects, the obtainment of patient’s consent and the gratification of the doctor-patient relationship may represent crucial concerns when telemedicine is used [155].

Whether and to what extent telemedicine affected patient outcome in terms of blood count control, symptom control and complications, remains to be clarified.


*During the COVID-19 pandemic, the integration of telemedicine and virtual care into the healthcare system emerged as an approach to maximize the efficiency of healthcare delivery by promoting social distancing, reducing face-to-face contacts and virus transmission and avoiding cancelling or postpone outpatient medical visits.*



*However, telemedicine seems to be more feasible and promising in ET and PV patients with stable disease who are not enrolled into investigational trials. In addition, the presence of a dedicated medical team that can guarantee a high quality of medical care and the cooperation of patients and caregivers may significantly improve the quality of telemedicine. Future studies are required to assess whether telemedicine may negatively affect disease control and MPN-related complications.*


## 8. Conclusions

The COVID-19 pandemic induced a rapid reorganization of healthcare facilities and greatly influenced the management of patients with cancer, who had to face a reduced possibility to perform laboratory tests and hospital visits; some specific therapies have been interrupted or postponed. MPNs and particularly MF patients may be at increased risk of severe SARS-CoV-2 infection. Differently, ET patients are at higher risk of thrombosis. There is uncertainty in the definition of the specific infectious risk related to their disease and to their chronic treatments. In addition, management of thrombotic and hemorrhagic risk may be challenging. The clinical evidence collected to date provides guidance for a proper management of MPN patients during the pandemic (summarized in Table 3). MPN patients should be encouraged to receive vaccination against COVID-19; the use of mRNA-based vaccines has proved safe and effective. The vaccine efficacy in patients with MPN, compared to the different specific therapies and the potential specific risks associated to the different vaccines, remain to be defined.

Overall, the experience gained in this pandemic year has also represented an opportunity for improvement in some areas, including the development of telemedicine, the expansion of international cooperation and the acceleration of approval pathways and implementation of clinical trials.

## Figures and Tables

**Figure 1 cancers-13-04750-f001:**
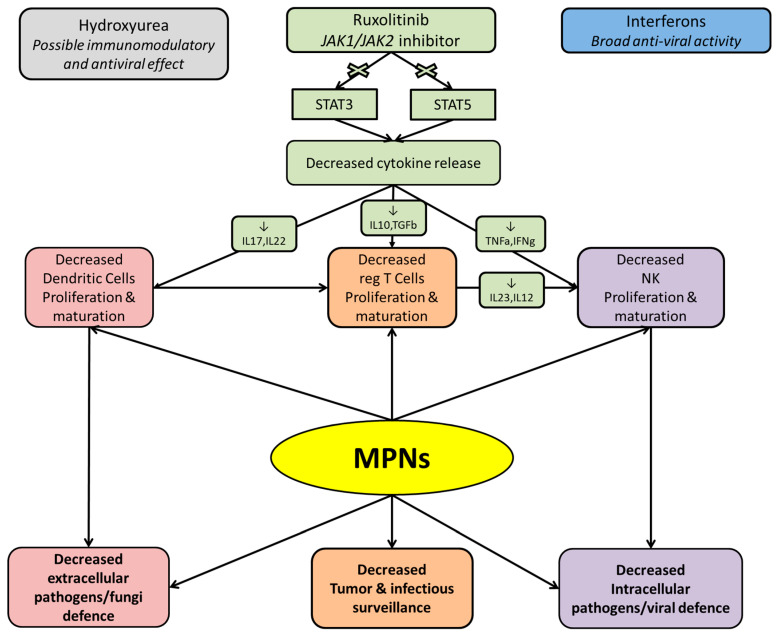
Pathways of immunodeficiency in patients with myeloproliferative neoplasms (MPN) and role of the JAK1/2 inhibitor ruxolitinib.

**Table 1 cancers-13-04750-t001:** Summary on the use of ruxolitinib and other *JAK2*-inhibitors in the treatment of SARS-CoV-2 infection. SoC: Standard of Care.

JAK2-Inhibitor	Locations	Study	Stage of COVID-19 Infection	Design	Therapy
Ruxolitinib	US, Argentina, Brazil, Colombia, France, Germany, Mexico, Peru, Russia, Spain, Turkey, UK	RUXCOVID, NCT04362137	COVID-19 associated cytokine storm requiring hospitalization	Phase 3, randomized, placebo-controlled	RUX 5 mg BID + SOC vs. PLACEBO + SOC
Ruxolitinib	UK	RAVEN, Eudract 2020-001777-71	COVID-19 associated cytokine storm requiring hospitalization	Phase 4, non-randomized, open label	RUX 5–20 mg BID
Ruxolitinib	Danmark	Eudract 2020-001459-42	Severe COVID-19 Infection	Phase 2, non-randomized, open label	RUX 5–20 mg BID
Ruxolitinib	Germany	RuXoCoil, Eudract 2020-001732-10 NCT04359290	Severe lung injury with ARDS	Single-arm, open label	RUX 5 mg BID
Ruxolitinib	Germany	RuxCoFlam, Eudract 2020-001481-11	Stage II/III COVID-19 with defined hyperinflammation	Phase 2, non-randomized, open label	RUX 5–20 mg BID
Ruxolitinib	China	ChiCTR-OPN-2000029580	Severe COVID-19 Infection	Single blind Randomized Controlled	RUX 5 mg BID in combination with mesenchymal stem cells vs. SOC
Ruxolitinib	US, Russian federation	RUXCOVID-DEVENT, NCT04377620	Severe lung injury with ARDS	Phase 3, randomized, placebo-controlled	RUX 5 mg BID + SOC vs. RUX 15mg BID + SOC vs. PLACEBO + SOC
Ruxolitinib	France	JAKINCOV, Eudract 2020-001963-10, NCT04366232	Severe COVID-19 Infection	Phase 2, randomized, open label	Anakinra 300 mg IV + RUX 5 mg BID vs. Anakinra 300 mg IV+ PLACEBO
Ruxolitinib	UK	MATIS, NCT04581954	mild or moderate COVID-19 pneumonia	multi-arm, multi-stage, randomised controlled trial	RUX 10mg BID Day 1–7 and 5 mg BID Day 8–14 vs. FOSTAMATINIB 150 mg BID Day 1–7 and 100 mg BID Day 8–14 vs. SOC
Pacritinib	US	PRE-VENT, NCT04404361	COVID-19 associated cytokine storm requiring hospitalization	Phase 3, randomized, double-blind, placebo-controlled	Pacritinib + SOX vs. Placebo + SOC

**Table 3 cancers-13-04750-t003:** Suggested strategies in the management of chronic myeloproliferative neoplasms under COVID-19 pandemic.

Disease	Diagnostic Procedures	Initial Therapy	Intolerant/Resistant Patients	Confirmed COVID-19
Polycythemia Vera	All patients should receive a 2016WHO-defined diagnosis. A delay of BM biopsy may be considered if clinical/laboratory parameters are diagnostic for PV	Anti COVID-19 vaccination is indicated Patients do not need be tested for COVID-19 prior to initiation of therapy. Antiplatelet agents according to standard indications. If newly diagnosed indication for oral anticoagulation, DOAC instead of VKA may be appropriate. In patients treated with phlebotomy only, the hematocrit threshold should be kept <45%Cytoreduction should be started in all patients at high thrombotic risk. The cytoreductive agent should be chosen on a case-by-case evaluation	There is no contraindication of switching to a second line cytoreductive agent in case of intolerance or resistance. The start of ruxolitinib should not be delayed	For non-severe COVID-19 infection, interruption of cytoreductive agents or ruxolitinib is not recommended. For severe COVID-19 infection, dose reduction or interruption of cytoreductive agents should be based on complete blood count evaluation. The interruption of ruxolitinib during COVID-19 should be discouraged, but discussed case by case Caution should be taken with the drug-drug interactions between treatment of COVID-19 and ruxolitinib. Switch to LMWH may be suggested in patients on anticoagulation. The use of LMWH is recommended in all hospitalized cases, after evaluation of the hemorrhagic risk Aspirin should not be discontinued in the patients with a history of arterial thrombosis
Essential Thrombocythemia	All patients should receive a 2016WHO-defined diagnosis. A delay of BM biopsy after the resolution of the pandemic may be considered if a MPN driver mutation or another clonal marker is present and clinical/laboratory parameters are in line with ET	Anti COVID-19 vaccination is indicated COVID-19 swab/serology is not required but it may be suggested prior to initiation of therapy Antiplatelet agents according to standard indications If newly diagnosed indication for oral anticoagulation, DOAC instead of VKA may be appropriate. Cytoreduction should be started in all patients at high thrombotic risk. The cytoreductive agent should be chosen on a case-by-case evaluation	There is no contraindication of switching to a second line cytoreductive agent in case of intolerance or resistance	For non-severe COVID-19, interruption of cytoreductive agents is not recommended. For severe COVID-19, dose reduction or interruption of cytoreductive agents should be based on complete blood count evaluation. Switch to LMWH may be suggested in patients on anticoagulation The use of LMWH is recommended in all hospitalized cases, after evaluation of the hemorrhagic risk Aspirin should not be discontinued in the patients with a history of arterial thrombosis
Myelofibrosis	All patients should receive a 2016 WHO-defined diagnosisA delay of BM biopsy after the resolution of the pandemic should be discouraged	Anti COVID-19 vaccination is indicated Patients do not need be tested for COVID-19 prior to initiation of therapy. The initiation of ruxolitinib should not be delayed if clinically needed Hydroxyurea can be started according to clinical need Initiation of anti-anemia therapy should be started to reduce the need of RBC transfusions	There is no contraindication of switching to cytoreductive agents/fedratinib Splenectomy should not be delayed if indicated since there are no data indicating an increased risk of COVID-19 infection/complication. The delay could exacerbate abdominal symptoms and delay ASCT. Pre-splenectomy vaccine prophylaxis is recommended. The indication and timing of ASCT are based on disease status.	The interruption of ruxolitinib during COVID-19 infection should be discouraged but discussed case by case Caution should be taken with the drug-drug interactions between treatment of COVID-19 and ruxolitinib Switch to LMWH may be suggested in patients on anticoagulation. The use of LMWH is recommended in all hospitalized cases, after evaluation of the hemorrhagic risk Aspirin should not be discontinued in the patients with a history of arterial thrombosis In patients with MF and thrombocytopenia MF-related, special attention should be paid to the risk/benefit balance associated with the antithrombotic prophylaxis

ASCT: allogeneic stem cell transplantation. BM: bone marrow. DOAC: direct oral anticoagulants. LMWH: low-molecular weight heparin. RBC: red blood cells. VKA: vitamin K antagonists. In MPN patients with severe COVID-19, that present CVRF but no history of arterial thrombosis, LMWH is recommended, while aspirin can be discontinued.

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
