# Peer review of "Philadelphia-Negative Chronic Myeloproliferative Neoplasms during the COVID-19 Pandemic: Challenges and Future Scenarios"

_cancers, 2021, doi:10.3390/cancers13194750_

Round 1

Reviewer 1 Report

General comments

Palandri at al. present a comprehensive review of the current literature on what is known about COVID-19 infections in patients with Philadelphia-negative chronic myeloproliferative neoplasms. Generally, this paper is well written expect but has some language editing needs. The paper is well-structured and has the potential to become a valuable reference work for physicians and researchers not only in the MPN field but also for the general hematology field as well as those interested in the impact of COVID19 on patients with malignancies. Obviously, the authors have put a lot of effort into reviewing the literature, as reflected by the large numbers of cited papers. The result is a comprehensive and high-quality work.

However, in my opinion there are some issues that need to be addressed in a revised version of the manuscript. The points are outlined in detail below:

Specific comments

  • Abstract: the impact of COVID-19 infection on cardio/vascular complication rates is an important aspect that should be summarized in the abstract. Also, the authors might consider to add a statement such as “ Based on the published data, strategies for the management of chronic MPNs are provided “ and already here refer to Table 3, which I believe will be the most useful piece of information for the large number of physicians taking care of MPN patients.
  • Page 1, line 41/42: percentages add up to more than 100%, please correct
  • Page3, lines 115-116: could an alternative explanation for the observed low incidence of COVID-19 infections in this relatively small patient population be random variations in infection rates? Not sure if this study “demonstrated” that adherence to basic rules prevented infections. Reference 35 citation is incomplete (reference list).
  • Page 5, line 221: does this recommendation (“prophylactic doses of LMWH”) relate to all hospitalized MPN patients or only to those with COVID-19 infection.
  • Page 6, line 228-232: Does this statement also hold true for hospitalized patients receiving heparin or only for the outpatient setting? The answer is provided later (lines 244/45) and thus this paragraph might benefit from restructuring.
  • Page 5- 6, lines 273 and following: this paragraph can be shortened considerably, especially with regard to the more general information on HU and IFN treatment that is not related to COVID-19
  • Page 7-8, lines 325 and following: see comment on general information on HU and IFN
  • Page 8, line 338: “JAK1 inhibition”. Is this correct?
  • Page 8 – 9, lines 382 and following: all information not relevant to COVID-19 can be shortened considerably
  • Page 9, line 397: please provide the appropriate reference(s)
  • Page 12, lines 460 and following: the description of the value of telemedicine can be shorted considerably. The most important question is if telemedicine affected patient outcome (blood count control, symptom control, complications, etc.), and this should be discussed.
  • Page 13, line 525: please provide the references showing “proof” that mRNA vaccines are most effective. Are there any randomized comparative studies done?
  • Table 3, ET part: ”delay of bone marrow biopsy”, consider change “clinical/laboratory parameters are suggestive” to “a MPN driver mutation or another clonal marker is present and clinical/laboratory parameters are in line with ET ”
  • Table 3, ET part, severe COVID-19 : I think also dose-reduction and not only treatment interruption is an option, depending on blood counts of course
  • Table 3, PV part: is the target value for HCT correct or should it be < 45%
  • Table 3, PV part, Aspirin use: what about patients with cardiac risk factors? Stop Aspirin or continue?
  • Table 3, MF part, splenectomy: Splenectomy is performed very rarely nowadays, is there any data on splenectomy in MPN and COVID-19 that would justify this statement. If yes, please add to the text and provide the reference.
  • The text is in need for final language editing before publication

Author Response

RESPONSES TO REVIEWER 1

General comments

Palandri at al. present a comprehensive review of the current literature on what is known about COVID-19 infections in patients with Philadelphia-negative chronic myeloproliferative neoplasms. Generally, this paper is well written expect but has some language editing needs. The paper is well-structured and has the potential to become a valuable reference work for physicians and researchers not only in the MPN field but also for the general hematology field as well as those interested in the impact of COVID19 on patients with malignancies. Obviously, the authors have put a lot of effort into reviewing the literature, as reflected by the large numbers of cited papers. The result is a comprehensive and high-quality work.

However, in my opinion there are some issues that need to be addressed in a revised version of the manuscript. The points are outlined in detail below:

Specific comments

  • Abstract: the impact of COVID-19 infection on cardio/vascular complication rates is an important aspect that should be summarized in the abstract. Also, the authors might consider to add a statement such as “ Based on the published data, strategies for the management of chronic MPNs are provided “ and already here refer to Table 3, which I believe will be the most useful piece of information for the large number of physicians taking care of MPN patients.

Ok, the abstract has been modified accordingly.

  • Page 1, line 41/42: percentages add up to more than 100%, please correct

Ok thanks, the percentages have been corrected.

  • Page3, lines 115-116: could an alternative explanation for the observed low incidence of COVID-19 infections in this relatively small patient population be random variations in infection rates? Not sure if this study “demonstrated” that adherence to basic rules prevented infections. Reference 35 citation is incomplete (reference list).

Ok, we agree with the suggestion of the Reviewer and we have added a specific comment on this. Reference 35 has been completed (the article is Online ahead of print.)

  • Page 5, line 221: does this recommendation (“prophylactic doses of LMWH”) relate to all hospitalized MPN patients or only to those with COVID-19 infection.

Only to those with COVID-19 infection. This has been clarified in the text.

  • Page 6, line 228-232: Does this statement also hold true for hospitalized patients receiving heparin or only for the outpatient setting? The answer is provided later (lines 244/45) and thus this paragraph might benefit from restructuring.

We thank the Reviewer for this suggestion. We have restructured the paragraph accordingly. Please note that the last sentences of each paragraph represent the response to the question that the paragraph aims to address. Therefore, the repetition of some parts of the above text is intentional and necessary. This has been now highlighted through the use of bold italics.

  • Page 5- 6, lines 273 and following: this paragraph can be shortened considerably, especially with regard to the more general information on HU and IFN treatment that is not related to COVID-19

Ok, we have shortened this paragraph.

  • Page 7-8, lines 325 and following: see comment on general information on HU and IFN

Ok, we have shortened this paragraph.

  • Page 8, line 338: “JAK1 inhibition”. Is this correct?

Thanks. It has been corrected to JAK1/JAK2.

  • Page 8 – 9, lines 382 and following: all information not relevant to COVID-19 can be shortened considerably

Ok, we have shortened this paragraph

  • Page 9, line 397: please provide the appropriate reference(s)

Ok. The appropriate reference has been provided.

  • Page 12, lines 460 and following: the description of the value of telemedicine can be shorted considerably. The most important question is if telemedicine affected patient outcome (blood count control, symptom control, complications, etc.), and this should be discussed.

Ok, we have shortened this paragraph and added a comment on this important issue.

  • Page 13, line 525: please provide the references showing “proof” that mRNA vaccines are most effective. Are there any randomized comparative studies done?

We acknowledge that there is no randomized trial proving superiority of a vaccine over another. Therefore, we have modified the sentence accordingly.

  • Table 3, ET part: ”delay of bone marrow biopsy”, consider change “clinical/laboratory parameters are suggestive” to “a MPN driver mutation or another clonal marker is present and clinical/laboratory parameters are in line with ET ”

Ok. It has been rephrased as suggested.

  • Table 3, ET part, severe COVID-19 : I think also dose-reduction and not only treatment interruption is an option, depending on blood counts of course

Ok. Dose reduction has been added.

  • Table 3, PV part: is the target value for HCT correct or should it be < 45%

Yes, thanks. “at 45%” has been corrected to “<45%”

  • Table 3, PV part, Aspirin use: what about patients with cardiac risk factors? Stop Aspirin or continue?

Thanks to the Reviewer for this comment, on the basis of which we thought it appropriate to unify the indications for antithrombotic prophylaxis to all three diseases (ET, PV, MF) and to point out in the text that pre-PMF has a similar thrombotic risk to ET. In addition, we added a caveat for patients with MF and thrombocytopenia. In patients with CVRF but no history of arterial thrombosis, LMWH is recommended in case of severe COVID-19, while aspirin can be discontinued. We hope that in this form the indications regarding anticoagulant/antiplatelet therapy may be clearer.

  • Table 3, MF part, splenectomy: Splenectomy is performed very rarely nowadays, is there any data on splenectomy in MPN and COVID-19 that would justify this statement. If yes, please add to the text and provide the reference.

We thank the Reviewer for this comment. Splenectomy is usually performed as a bridge to ASCT, and its timing is therefore very important. Of course, there is no clinical report on splenectomy in MF during the pandemic. Here, we wanted to provide expert recommendation that if the patient is in need of splenectomy (mostly, to undergo ASCT), surgery should not be delayed due to the pandemic.

Reviewer 2 Report

In this well-written review, the authors report on the COVID-19 challenges in patients with MPN focusing on the risk profile, therapy management, vaccination strategies, and telemedicine.

Major points:

Line 76-82: The authors write: Infections are frequently reported in MPNs, representing the ultimate cause of death in approximately 10% of patients [17,18]. Infections are primarily bacterial (78%) but viral (11%) and fungal (2%) infections can also develop[17]. Infectious risk in MPNs is mainly caused by deregulation of key mediators of the immune system. In particular, monocytes/macrophages, T cells, natural killers, and myeloid-derived suppressor cells are often characterized by numerical and/or functional abnormalities [19-24]. The most recent work should be cited in the text: Landtblom AR, Andersson TM, Dickman PW, Smedby KE, Eloranta S, Batyrbekova N, Samuelsson J, Björkholm M, Hultcrantz M. Risk of infections in patients with myeloproliferative neoplasms-a population-based cohort study of 8363 patients. Leukemia. 2021 Feb;35(2):476-484. doi: 10.1038/s41375-020-0909-7. Epub 2020 Jun 16. PMID: 32546727; PMCID: PMC7738400

Line 151-155: The study from Spain observed a lower mortality in MPN patients. Is this compared to a healthy population or compared to AML and non-Hodgkins lymphoma, the latter not surprising ?

Line 180: The likelihood of having COVID-19 seems higher in MPNs. you should delete “especially in ET” because the PMF group was not included in the analysis of the 73 million patients from the US.

Line 273-288 (and figure 1): Regarding the use of hydroxyurea, it has been shown that it has an antiviral effect in HIV and an immunomodulatory effect in sickle cell anemia. The authors should elaborate on this in regard to its potential use in COVID-19 afflicted MPN patients (Hasselbalch et al, Cytokine and Growth Factor Reviews 60 (2021) 28–45. F. Lori, J. Lisziewicz, Hydroxyurea: overview of clinical data and antiretroviral and immunomodulatory effects, Antivir Ther. 4 (Suppl 3) (1999) 101–108. C.C. Guarda, P.S.M. Silveira-Mattos, S.C.M.A. Yahou´ed´ehou, et al., Hydroxyurea alters circulating monocyte subsets and dampens its inflammatory potential in sickle cell anemia patients, Sci. Rep. 9 (2019) 14829).

Line 293: The important IFN paper by Hasselbalch HC. A new era for IFN-α in the treatment of Philadelphia-negative chronic myeloproliferative neoplasms. Expert Rev Hematol. 2011 Dec;4(6):637-55 is missing in this long list of cited papers and should be added.

Line 386-388: possibly because of a too low dose of ruxo.

Section 7 from line 460: This section of telemedicine is heavily biased towards all the pros of using telemedicine with only three lines discussing cons. The authors should include more space to discuss the drawbacks of the lack of physical interaction. In this context, it is important to state that other disease symptoms may be overlooked when adhering to telemedicine compared to having the “in person” close contact to the doctor.

To complete this review, the authors should discuss and cite the comprehensive review by Hasselbalch et al, Cytokine and Growth Factor Reviews 60 (2021) 28-45 (online April 2021), which addresses so many aspects of COVID-19 in MPNs regarding treatment with hydroxyurea, ruxolitinib and/or interferon.

Minor points:

Line 16: delete “about”

Line 104: delete “in” (written twice)

Line 108: replace “resulted” with “were”

Line 121: when they had the opportunity  

Line 199: delete “and”

Line 206: low molecular weight heparin (LMWH)

Line 280: delete comma before “that”

Table 2: In the row with Moderna and systemic adverse advents, you have written 54.9% as both 1st and 2nd and 79.4% as 2nd. Only one of the percentages can be 2nd.

Author Response

RESPONSES TO REVIEWER 2

In this well-written review, the authors report on the COVID-19 challenges in patients with MPN focusing on the risk profile, therapy management, vaccination strategies, and telemedicine.

Major points:

  • Line 76-82: The authors write: Infections are frequently reported in MPNs, representing the ultimate cause of death in approximately 10% of patients [17,18]. Infections are primarily bacterial (78%) but viral (11%) and fungal (2%) infections can also develop[17]. Infectious risk in MPNs is mainly caused by deregulation of key mediators of the immune system. In particular, monocytes/macrophages, T cells, natural killers, and myeloid-derived suppressor cells are often characterized by numerical and/or functional abnormalities [19-24]. The most recent work should be cited in the text: Landtblom AR, Andersson TM, Dickman PW, Smedby KE, Eloranta S, Batyrbekova N, Samuelsson J, Björkholm M, Hultcrantz M. Risk of infections in patients with myeloproliferative neoplasms-a population-based cohort study of 8363 patients. 2021 Feb;35(2):476-484. doi: 10.1038/s41375-020-0909-7. Epub 2020 Jun 16. PMID: 32546727; PMCID: PMC7738400

Yes, we acknowledge that the paper cited by the Reviewer is of utmost importance. The article has been added to the references.

As agreed with Assistant Editor Dr. Eden Xiao, new references have been added numerically at the bottom of the list (this reference is temporarily n. 167) and will be reordered at a later time by the Journal.

  • Line 151-155: The study from Spain observed a lower mortality in MPN patients. Is this compared to a healthy population or compared to AML and non-Hodgkins lymphoma, the latter not surprising?

It is compared to AML and NHL, indeed not much surprising. This clarification has been added in the text.

  • Line 180: The likelihood of having COVID-19 seems higher in MPNs. you should delete “especially in ET” because the PMF group was not included in the analysis of the 73 million patients from the US.

Ok. It has been removed.

  • Line 273-288 (and figure 1): Regarding the use of hydroxyurea, it has been shown that it has an antiviral effect in HIV and an immunomodulatory effect in sickle cell anemia. The authors should elaborate on this in regard to its potential use in COVID-19 afflicted MPN patients (Hasselbalch et al, Cytokine and Growth Factor Reviews 60 (2021) 28–45. F. Lori, J. Lisziewicz, Hydroxyurea: overview of clinical data and antiretroviral and immunomodulatory effects, Antivir Ther. 4 (Suppl 3) (1999) 101–108. C.C. Guarda, P.S.M. Silveira-Mattos, S.C.M.A. Yahou´ed´ehou, et al., Hydroxyurea alters circulating monocyte subsets and dampens its inflammatory potential in sickle cell anemia patients, Sci. 9 (2019) 14829).

Ok. We thank the Reviewer for this important indication. We have followed the suggestion and these articles have been added to the references. Also, we have update Figure 1 accordingly.

As agreed with Assistant Editor Dr. Eden Xiao, new references have been added numerically at the bottom of the list (these references are temporarily n. 170 and 172) and will be reordered at a later time by the Journal.

  • Line 293: The important IFN paper by Hasselbalch HC. A new era for IFN-α in the treatment of Philadelphia-negative chronic myeloproliferative neoplasms. Expert Rev Hematol. 2011 Dec;4(6):637-55 is missing in this long list of cited papers and should be added.

Ok, the article has been added to the references.

As agreed with Assistant Editor Dr. Eden Xiao, new references have been added numerically at the bottom of the list (this reference is temporarily n. 174) and will be reordered at a later time by the Journal.

  • Line 386-388: possibly because of a too low dose of ruxo.

In order to fulfill reviewer #1's requests, this sentence has been removed from the text.

  • Section 7 from line 460: This section of telemedicine is heavily biased towards all the pros of using telemedicine with only three lines discussing cons. The authors should include more space to discuss the drawbacks of the lack of physical interaction. In this context, it is important to state that other disease symptoms may be overlooked when adhering to telemedicine compared to having the “in person” close contact to the doctor.

We acknowledge the validity of this observation. Therefore, we have elaborated more fully on the disadvantages of telemedicine.

  • To complete this review, the authors should discuss and cite the comprehensive review by Hasselbalch et al, Cytokine and Growth Factor Reviews 60 (2021) 28-45 (online April 2021), which addresses so many aspects of COVID-19 in MPNs regarding treatment with hydroxyurea, ruxolitinib and/or interferon.

We thank the Reviewer for this suggestion. The article has been added to the references.

As agreed with Assistant Editor Dr. Eden Xiao, new references have been added numerically at the bottom of the list (this reference is temporarily n. 170)  and will be reordered at a later time by the Journal.

Minor points:

  • Line 16: delete “about”

Ok.

  • Line 104: delete “in” (written twice)

Ok.

  • Line 108: replace “resulted” with “were”

Ok.

  • Line 121: when they had the opportunity  

Ok.

  • Line 199: delete “and”

Ok.

  • Line 206: low molecular weight heparin (LMWH)

Ok.

  • Line 280: delete comma before “that”

Ok.

  • Table 2: In the row with Moderna and systemic adverse advents, you have written 54.9% as both 1st and 2nd and 79.4% as 2nd. Only one of the percentages can be 2nd.

Ok. Thank you for your corrections.

Reviewer 3 Report

In this Review Palandri et al., discusses the risk of MPN patients to be infected by SARS-CoV-2, the risk of MPN patients with COVID19 to develop severe disease, recommendations for MPN therapy initiation, continuation or switch in general in the context of the pandemic and specifically in COVID-affected patients. In addition, the authors also indicated vaccination strategies for MPN patients and the appropriateness of telemedical care in pandemic times and beyond. The manuscript is well-structured and covered most important points of concern. This work is highly timely and should be of interest to the community.

I have only minor comments, which I hope will help to improve the delivery of the messages to a broad audience.

1) The following corrections should be made in the text:

Line 108: … ruxolitinib therapy resulted associated with higher infectious… should be corrected to ’’ruxolitinib therapy associated with higher infectious’.

Line 112: In 271 MPN German patients … should be corrected to ‘’In 271 German MPN patients’.

Lines 133-135: … A nation-wide database of patient elec-133 tronic health records of 73 million patients in the US… should be corrected to ‘’ A nation-wide database of patient electronic health records of 73 million patients in the US was analyzed for COVID-19 and eight major types of hematologic malignancies (including 121,200 ET patients and 134 72,150 PV patients, PMF was not included in the analysis)’’.

Line 148: … heathy.. should be corrected to ‘’healthy’’…

Line 252: … and monitoring … should be corrected to ‘’and to monitor’’.

Line 282: … grade 3 neutropenia, with no grade 4 events and reversible after drug temporary … ‘’grade 3 neutropenia without grade 4 events, which was reversible after drug temporary’’.

Line 308: … diod not changed .. should be corrected to ‘’did not change’’.

Lines 358-359: … Overall, most (79.8%) of the hematologists believed that ruxolitinib had no negative effect on COVID-19 infection, while 10.1% believed that…

Line 387: … since did.. should be corrected to ‘’since it did’’.

Lines 394-396: should be changed to ‘’However, a dose reduction of ruxolitinib may be temporarily performed to reduce drug-drug interactions worsening hematology parameters or patients’ clinical status’.

Lines 405-406: … should be changed to ‘’The SARS-CoV-2 spike (S) surface glycoprotein is a large highly antigenic type I’’…

Table 2, first and second row: …to be stored at…

Table 2, third and fourth row: highly stable

Lines 484-485: …hematologists treating MPN patients during the COVID-19 pandemic, particularly for patients…

Table 3 (Polycytemia vera and myelofibrosis, initial therapy): Patients do not need to be tested…

2) Some abbreviations, which are not mentioned in the main text should be indicated:

Line 124: Ph- (Philadelphia negative)

Line 206: LMWH (low-molecular weight heparin)

Line 224: VTE (Venous thromboembolism)

Line 272: Hct (hematocrit)

3) It is not clear what the authors refer to by mentioning ‘’drug-drug interactions during anticoagulant therapy (Line 242) and ruxolitinib treatment’’ (Line 394). The authors should specify.

Author Response

RESPONSES TO REVIEWER 3

In this Review Palandri et al., discusses the risk of MPN patients to be infected by SARS-CoV-2, the risk of MPN patients with COVID19 to develop severe disease, recommendations for MPN therapy initiation, continuation or switch in general in the context of the pandemic and specifically in COVID-affected patients. In addition, the authors also indicated vaccination strategies for MPN patients and the appropriateness of telemedical care in pandemic times and beyond. The manuscript is well-structured and covered most important points of concern. This work is highly timely and should be of interest to the community.

I have only minor comments, which I hope will help to improve the delivery of the messages to a broad audience.

1) The following corrections should be made in the text:

  • Line 108: … ruxolitinib therapy resulted associated with higher infectious… should be corrected to ’’ruxolitinib therapy associated with higher infectious’.

As requested by another reviewer, we have already rephrased to “ruxolitinib therapy were associated with higher infectious”. Thank you.

  • Line 112: In 271 MPN German patients … should be corrected to ‘’In 271 German MPN patients’.

Ok. It has been corrected.

  • Lines 133-135: … A nation-wide database of patient electronic health records of 73 million patients in the US… should be corrected to ‘’ A nation-wide database of patient electronic health records of 73 million patients in the US was analyzed for COVID-19 and eight major types of hematologic malignancies (including 121,200 ET patients and 134 72,150 PV patients, PMF was not included in the analysis)’’.

Ok.

  • Line 148: … heathy.. should be corrected to ‘’healthy’’…

Ok.

  • Line 252: … and monitoring … should be corrected to ‘’and to monitor’’.

Ok.

  • Line 282: … grade 3 neutropenia, with no grade 4 events and reversible after drug temporary … ‘’grade 3 neutropenia without grade 4 events, which was reversible after drug temporary’’.

Ok.

  • Line 308: … diod not changed .. should be corrected to ‘’did not change’’.

Ok.

  • Lines 358-359: … Overall, most (79.8%) of the hematologists believed that ruxolitinib had no negative effect on COVID-19 infection, while 10.1% believed that…

Ok.

  • Line 387: … since did.. should be corrected to ‘’since it did’’.

Ok.

  • Lines 394-396: should be changed to ‘’However, a dose reduction of ruxolitinib may be temporarily performed to reduce drug-drug interactions worsening hematology parameters or patients’ clinical status’.

Ok.

  • Lines 405-406: … should be changed to ‘’The SARS-CoV-2 spike (S) surface glycoprotein is a large highly antigenic type I’’…

Ok.

  • Table 2, first and second row: …to be stored at…

Ok.

  • Table 2, third and fourth row: highly stable

Ok.

  • Lines 484-485: …hematologists treating MPN patients during the COVID-19 pandemic, particularly for patients…

Ok.

  • Table 3 (Polycytemia vera and myelofibrosis, initial therapy): Patients do not need to be tested…

Ok, thank you for your corrections.

2) Some abbreviations, which are not mentioned in the main text should be indicated:

  • Line 124: Ph- (Philadelphia negative)
  • Line 206: LMWH (low-molecular weight heparin)
  • Line 224: VTE (Venous thromboembolism)
  • Line 272: Hct (hematocrit)

Ok. They are now indicated respectively al lines 73, 214, 197 and 265.

3) It is not clear what the authors refer to by mentioning ‘’drug-drug interactions during anticoagulant therapy (Line 242) and ruxolitinib treatment’’ (Line 394). The authors should specify.

Ok. We referred to possible metabolic interactions on liver cytochromes, such as CYP2C9 and CYP3A4, between RUX and anticoagulants and between RUX and antiviral agents against COVID-19. This point has been specified.

Round 2

Reviewer 2 Report

No further comments.